# Impact of hysterectomy on opioid use in patients with adenomyosis: A nationwide register study

Malin Brunes[1,2]*, Gudny Jonsdottir[3,4], Marion Ek[1,2], Helena Kopp Kallner[3,4], Klara Hasselrot[3,4]

**1** Division of Obstetrics and Gynecology, Södersjukhuset University Hospital, Stockholm, Sweden,
**2** Department of Clinical Sciences and Education, Södersjukhuset University Hospital, Karolinska Institutet, Stockholm, Sweden, **3** Department of Obstetrics and Gynecology, Danderyd Hospital, Stockholm, Sweden, **4** Department of Clinical Science at Danderyd Hospital, Division of Obstetrics and Gynecology, Karolinska Institutet, Stockholm, Sweden

* malin.brunes@regionstockholm.se

**Data Availability Statement:** The data in the study are pseudonymized (coded) personal data, and GDPR prohibits us from sharing this completely open. Data is available upon request, and requests

## Abstract

### Introduction

Dysmenorrhea and heavy menstrual bleeding are the most common symptoms in adenomyosis, in addition to infertility and chronic pelvic pain. Hysterectomy is a common treatment for adenomyosis symptoms with curative effect on heavy menstrual bleeding but with less studied effects on pain reduction.

### Material and methods

This is a nationwide retrospective register-based cohort study including all hysterectomized patients with pathology-verified adenomyosis between 1 January 2012 and 31 December 2015 with a long-term follow-up three years pre- and three years postoperatively. Two national registers were linked in order to describe the primary outcome of proportion with use of opioids among hysterectomized patients with and without adenomyosis pre- and postoperatively. Logistic and multinomial regression models were used.

### Results

A total of 2,228 (15%) patients had pathology-verified adenomyosis. Overall opioid use was 18.6% and 21.1% three years before and three years after surgery, respectively. Results showed a significantly increased risk of opioid use three years after hysterectomy in patients with preoperative use of opioids and antidepressants (adjusted Odds Ratio (aOR) 5.7, 95% Confidence Interval (CI) 4.5–7.2 and aOR 1.4, 95% CI 1.1–1.8). The risk of needing long-term opioids was higher among patients with smaller uteri (<300g, aOR 2.8, 95% CI 1.7–4.7) compared to women with uterine sizes ≥600g.

### Conclusions

Hysterectomy does not reduce opioid use among adenomyosis patients in long-term follow-up, although the subjective reduction of pain was not investigated in this study. Women with

for access to the data can be put to our Research Data Office (rdo@ki.se) at Karolinska Institutet.

**Funding:** M.E. received grant from Intuitive surgical grant number IS:5256675 The funders did not play any role in the study design, data collection and analysis, decision to publish, or preparation of the manuscript.

**Competing interests:** Malin Brunes have received honoraria from Intuitive Surgical for three lectures and proctoring. This does not alter our adherence to PLOS ONE policies on sharing data and materials.

**Abbreviations:** a, adjusted; ASA, The American Society of Anesthesiologists classification; ATC, Anatomical Therapeutic Chemical Classification System; BMI, body mass index; c, crude; CI, confidence interval; DDD, World Health Organization Defined Daily Dose; HMB, heavy menstrual bleeding; ICD-10, International Statistical Classification of Diseases and Related Health Problems; IRR, incidence-rate ratio; OR, odds ratio; SPDR, Swedish Prescribed Drug register.

preoperative use of opioids/antidepressants and uterine size <300g are at increased risk for chronic opioid use.

## Introduction

Although described by pathologist von Rokitansky in 1860 [1], more than 160 years later significant gaps remain in understanding adenomyosis and the best treatment modalities. The disease can cause substantial suffering, with dysmenorrhea and menorrhagia as the most common symptoms, in addition to infertility and chronic pelvic pain [2, 3].

Due to increasing use of hormonal treatment as a first line medication of heavy menstrual bleeding (HMB)/dysmenorrhea, numbers of hysterectomies due to these indications have declined in many countries [4, 5]. Although hysterectomy is curative for HMB, the effect on adenomyosis-related pain symptoms is less well understood. Large prospective trials concerning pain outcomes in adenomyosis patients after surgical interventions are lacking, and a randomized controlled trial regarding hysterectomy versus uterine artery embolization (QUESTA trial) is still ongoing [6].

Previously it has been shown that 7.1% of hysterectomy patients in Sweden used opioids three years after surgery, where adenomyosis correlated to long-term post-operative opioid treatment to a higher extent than endometriosis [7]. Opioid use as a proxy for moderate-severe pain has previously been established in patients with chronic pelvic pain [8]. The most commonly used opioids in Sweden from 2012 to 2015 were oxycodone, tramadol and paracetamol combined with codeine.

In this epidemiological study of the Swedish population, we aimed to study the long-term consequences of hysterectomy on opioid consumption, as a proxy for moderate to severe pain. Our secondary aim was to identify predictive factors for continued opioid use after hysterectomy.

## Materials and methods

The databases used have been described previously by Brunes [7], briefly summarized as follows: The Swedish National Quality Register of Gynecological Surgery contains data on gynecological surgery performed in Sweden, and 93% (52 of 56) of gynecological departments reported to the register during the study period. The register contains information on demographics as well as pre- intra- and postoperative data [9]. All data is prospectively collected and is registered online both by the patients (pre- and post-operative symptoms, as well as postoperative adverse effects) and by the operating surgeon (clinical exams, ultrasound and lab results pre-operatively, in addition to a complete report of the surgery performed). Main indication for surgery is reported both by the responsible physician (as an International Statistical Classification of Diseases and Related Health Problems ICD-10 code) as well as by the patient (according to symptoms). All surgical specimens are investigated by pathologists, and the subsequent results are reported to the register for correct anatomical-histological diagnosis. In Sweden all uteri with suspected benign disease are examined macroscopically by the pathologist, followed by 2 biopsies from cervix uteri and 4 biopsies from corpus uteri, and additional biopsies from myomas >5cm. From macroscopically suspected areas (for example, overt adenomyosis) additional biopsies are taken.

In Sweden, all prescribed drugs retrieved by a patient from a pharmacy are registered in the Swedish Prescribed Drug Register (SPDR), with a coverage of 97–100%. Most drugs in Sweden

are prescribed by a physician, and a prescription is necessary in order to receive state-funded subsidized drugs. No opioid-containing drugs and only smaller packages for daily use of Non-Steroid Anti-Inflammatory Drugs (NSAIDs) and paracetamol can be bought over the counter without a prescription.

The study population consisted of patients having a total hysterectomy due to benign indications during four years; 1 January 2012–31 December 2015. Data from this population was linked to the SPDR using the unique national registration number assigned to all Swedish citizens. Data extraction from the drug register covered a period from 1 January 2009 until 31 December 2018, in order to cover a period from three years prior to and three years after surgery. Drugs were classified using the Anatomical Therapeutic Chemical Classification System (ATC) code. Three main groups were identified: analgesics (drugs containing opioids, other analgesics, and muscle-relaxants), psychoactive drugs (sedatives and sleeping pills and anti-depressants) and neuroactive drugs (neuroleptic drugs, antiepileptic drugs and psychostimulants) (S1 Table). ATC code N02A (opioids) includes, for example, morphine, oxycodone and tramadol, as well as combination drugs containing paracetamol, along with codeine.

Data were accessed for research purposes in 24/12/2019.

Patients with adenomyosis were identified through the ICD-10 code N80.0 from the register. Our primary objective was to assess pre- and postoperative variations in the prescription of opioids (defined as yes/no), as well as psychotropic and neuroactive medications, among women diagnosed with adenomyosis undergoing hysterectomy. Data on the proportion of patients using the studied medications was collected for each year of the study period. As a secondary outcome, in the opioid group, differences in the World Health Organization Defined Daily Dose (DDD) were analyzed yearly before and after hysterectomy (12). Additionally, we conducted a subgroup analysis, categorizing patients with adenomyosis into two subgroups: 1) no long-term use, comprising patients using postoperative opioids for 0–12 months (yes/no), and 2) long-term use, encompassing patients using opioids for more than 12 months up to three years post-hysterectomy (yes/no). Potential significant risk factors of long-term postoperative opioid use were based on previous research and clinical discussions:

a. Presence of pre-operative prescribed opioids, antidepressants and sedatives/sleeping pills [10]

b. Previous abdominal surgery

c. Surgical complications [10]

d. Abdominal hysterectomy

The code N80.0 is registered by the responsible gynecologist, based on either pre-surgical diagnosis by ultrasound or other imaging, or after a pathologist report.

Co-morbidities such as concomitant depression and chronic pain are not registered in the Swedish National Quality Register of Gynecological Surgery.

Baseline characteristics of patients included: age, body mass index (BMI), The American Society of Anesthesiologists classification (ASA), smoking, parity, gynecological comorbidity, uterine specimen size and pre-operative hormonal treatment.

The software Stata v 13.0 (StataCorp LLC) was used for statistical analysis. Baseline characteristics are presented as frequencies and proportions for categorical variables.

For the analyses of risk factors for long term use of opioids in Table 1 we used logistic or multinomial regression. Since some variables in the register had a higher rate of missing data, we conducted both a complete and a stepwise multivariable regression. In the stepwise logistic regression model, we excluded all variables with missing data over 25%.

For the categorical primary endpoints, namely, pre- and postoperative prescription of drugs (analgesics, psychoactive and neuroactive drugs) univariable logistic regression was used.

To compare mean DDD in prescription of drugs containing opioids negative binomial regression was performed. When comparing prescription of drugs before and after hysterectomy the results were not adjusted for confounders since this was the same individuals and thus self-controlled. To account for intraindividual dependence, the robust sandwich estimator for standard error was used. Results from the univariable regression model are presented as crude odds ratio (cOR), from the multivariable regression model as adjusted odds ratio (aOR) and from the negative binomial regression model as incidence-rate ratio IRR with a 95% confidence interval (CI). Statistical significance was set to $p < 0.05$.

The study was approved by the research ethics committee at Karolinska Institutet, Stockholm, Sweden (2018/190-31) and conforms to the STROBE guidelines for reporting observational studies (www.strobe-statement.org). The register obtained informed consent from patients, providing information that data may be used for research purposes. All data from the registers were anonymized before being accessed by researchers. The ethics committee waived the requirement for informed consent because the risk of privacy intrusion was considered low compared to the benefits of the research.

## Results

During the four years studied, 14,682 patients in Sweden underwent hysterectomy due to benign indications, (excluding indications due to obstetric complications and prolapse), and 2,228 of these patients had a post-operative verified diagnosis of adenomyosis, representing 15% of the investigated population (Fig 1). Before surgery, 86 of 2,228 patients (3.9%) had adenomyosis registered as main indication for surgery. Main registered indications for hysterectomy (on the day of surgery) in this dataset includes "menorraghia" (36%), "myoma" (18.9%), "dysplasia" (2.8%), "adnexal pathology" (2.7%), "endometriosis" (2.7%) and "dysmenorrhea" (1.5%), and the rate of missing data was 29.5%.

The primary outcome, namely the level of opioid use among adenomyosis patients, was 18.6% three years before surgery and 21.1% three years after (Fig 2).

This finding was substantiated by the secondary outcome in patients diagnosed with adenomyosis, a statistically significant increase in DDD opioids (IRR 1.1, 95% CI 1.0–1.3) was shown three years after, compared to three years before hysterectomy (S1 Fig).

The use of psycho- (OR 1.5, 95% CI 1.3–1.7) and neuroactive drugs (OR 1.6, 95% CI 1.2–2.1) increased in patients with adenomyosis three years after surgery (Fig 3).

In the sub-group of patients needing long-term (>one year) opioid analgesia (n = 641), the identified risk-factors prior to surgery were as follows: use of prescribed opioids and use of antidepressants one-year pre-surgery (aOR 5.7, 95% CI 4.5–7.2 and aOR 1.4, 95% CI 1.1–1.8). The risk of needing long-term opioids was higher among patients with smaller uteri (<300g, aOR 2.8, 95% CI 1.7–4.7) compared to women with uterine sizes ≥600g (Table 1). A total of 314 patients (49%) in the long-term opioid group were under 45 years old. However, age was not identified as a statistically significant risk factor for long-term opioid use.

Neither parity, BMI, smoking or previous abdominal surgery was significantly associated with long-term postoperative consumption of opioids, neither was the prevalence of endometriosis (15.4% in long-term users vs 15.9%) or the use of preoperative hormonal treatment (69.6% in long-term users vs 63.8%) (Table 1). In a subgroup analysis (S2 Fig) we could not find any significant differences in route of hysterectomy when comparing opioid use three years before to three years after hysterectomy (in total: 948 (43%) AH, 580 (26%) TLH/RTLH, 699 (31%) VH).

**Table 1. Baseline characteristics of the study population including analysis of risk-factors for need of long-term opioids in patients with adenomyosis (n = 2,228).**

| Baseline characteristics | [1] No long-term opioids (<one year) n = 1,587 | Long-term opioids (>one year) n = 641 | cOR (95% CI) | aOR1 (95% CI) | aOR2 (95% CI) |
|---|---|---|---|---|---|
| Age | | | | | |
| •< 45 | 912 (57.5) | 314 (49.0) | 1.5 (1.2–1.8) | 1.2 (0.9–1.6) | 1.1 (0.9–1.4) |
| •45–55 | 518 (32.6) | 268 (41.8) | Ref | Ref | Ref |
| •> 55 | 157 (9.9) | 59 (9.2) | 1.1 (0.8–1.5) | 1.0 (0.6–1.8) | 1.2 (0.8–1.7) |
| BMI | | | | | |
| •<25 | 423 (26.7) | 130 (20.3) | Ref | Ref | |
| •25–29 | 379 (23.9) | 143 (22.3) | 1.2 (0.9–1.6) | 1.1 (0.8–1.6) | |
| •>29 | 234 (14.7) | 112 (17.5) | 1.6 (1.2–2.1) | 1.3 (0.9–1.9) | |
| ASA | | | | | |
| •1–2 | 1,534 (96.7) | 616 (96.1) | Ref. | Ref | Ref |
| •3–4 | 29 (1.8) | 20 (3.1) | 1.7 (1.0–3.1) | 1.6 (0.6–4.4) | 1.5 (0.8–3.0) |
| Smoking | 154 (9.7) | 71 (11.1) | 1.3 (0.9–1.8) | 1.2 (0.8–1.7) | |
| Parity | | | | | |
| •0 | 93 (5.9) | 37 (5.8) | Ref. | Ref. | |
| •1–2 | 593 (37.4) | 202 (31.5) | 0.9 (0.6–1.3) | 0.9 (0.5–1.5) | |
| •≥3 | 329 (20.7) | 143 (22.3) | 1.1 (0.7–1.7) | 1.0 (0.6–1.6) | |
| Gynecological comorbidity | | | | 0.9 (0.6–1.2) | 1.0 (0.8–1.2) |
| •Myoma | 638 (40.2) | 196 (30.6) | 0.7 (0.5–0.8) | 0.7 (0.5–1.1) | 0.7 (0.5–1.0) |
| •Endometriosis | 252 (15.9) | 99 (15.4) | 1.0 (0.8–1.2) | | |
| Previous abdominal surgery | | | | | |
| | 668 (42.1) | 271 (42.3) | 1.3 (1.0–1.7) | 1.0 (0.7–1.4) | |
| Route of surgery | | | | Ref. | Ref |
| •AH | 714 (45.0) | 234 (36.5) | Ref | 1.1. (0.8–1.6) | 1.1. (0.8–1.5) |
| •TLH/RTLH | 379 (23.9) | 201 (31.4) | 1.6 (1.3–2.0) | 1.0 (0.7–1.4) | 1.0 (0.7–1.3) |
| •VH | 493 (31.1) | 206 (32.1) | 1.3 (1.0–1.6) | | |
| Specimen size (g) | | | | | |
| •< 300 | 1,063 (67.0) | 538 (83.9) | Ref | Ref | Ref |
| •300–599 | 290 (18.3) | 59 (9.2) | 0.4 (0.3–0.5) | 0.7 (0.4–1.1) | 0.5 (0.4–0.7) |
| •≥ 600 | 158 (10.0) | 22 (3.4) | 0.3 (0.2–0.4) | 0.3 (0.1–0.6) | 0.4 (0.2–0.6) |
| Specimen size (g) | | | | | |
| •< 300 | 1,063 (67.0) | 538 (83.9) | 3.6 (2.3–5.7) | 3.9 (1.8–8.5) | 2.8 (1.7–4.7) |
| •300–599 | 290 (18.3) | 59 (9.2) | 1.5 (0.9–2.5) | 2.6 (1.1–5.8) | 1.4 (0.8–1.7) |
| •≥ 600 | 158 (10.0) | 22 (3.4) | Ref | Ref | Ref |
| Preoperative hormonal treatment | | | | | |
| | 1,013 (63.8) | 446 (69.6) | 1.2 (0.8–1.7) | 1.0 (0.7–1.4) | 1.1 (0.9–1.4) |
| Perioperative complications | | | | | |
| | 53 (3.3) | 13 (2.0) | 0.6 (0.3–1.1) | 0.9 (0.4–2.1) | 0.7 (0.3–1.3) |
| Opioids 1-year preop | 203 (12.8) | 317 (49.5) | 6.7 (5.4–8.3) | 6.3 (4.5–8.7) | 5.7 (4.5–7.2) |
| Antidepressant 1-year preop | | | | | |
| | 295 (18.6) | 219 (34.2) | 2.3 (1.8–2.8) | 1.3 (0.9–1.8) | 1.4 (1.1–1.8) |
| Sedatives and sleeping pills 1-year preop | 308 (19.4) | 213 (33.2) | 2.1 (1.7–2.5) | 1.2 (0.9–1.8) | 1.2 (1.0–1.6) |

[1]Reference = Postoperative opioids (<12 months)

BMI, body mass index

Figures are frequencies (proportions).

cOR (95% CI) = crude Odds Ratio (95% Confidence Interval)

aOR1 (95%CI) = adjusted Odds Ratio (95% Confidence Interval) Complete analysis: all variables included

aOR2 (95%CI) = adjusted Odds Ratio (95% Confidence Interval) variables with missing data over 25% excluded; main indication, BMI, parity, smoking

To address possible characteristics of adenomyosis only patients (no concomitant endometriosis and/or myomas) vs adenomyosis as additional diagnosis, we performed a subgroup analysis (S3 Fig) showing no significant difference in opioid use between these two groups.

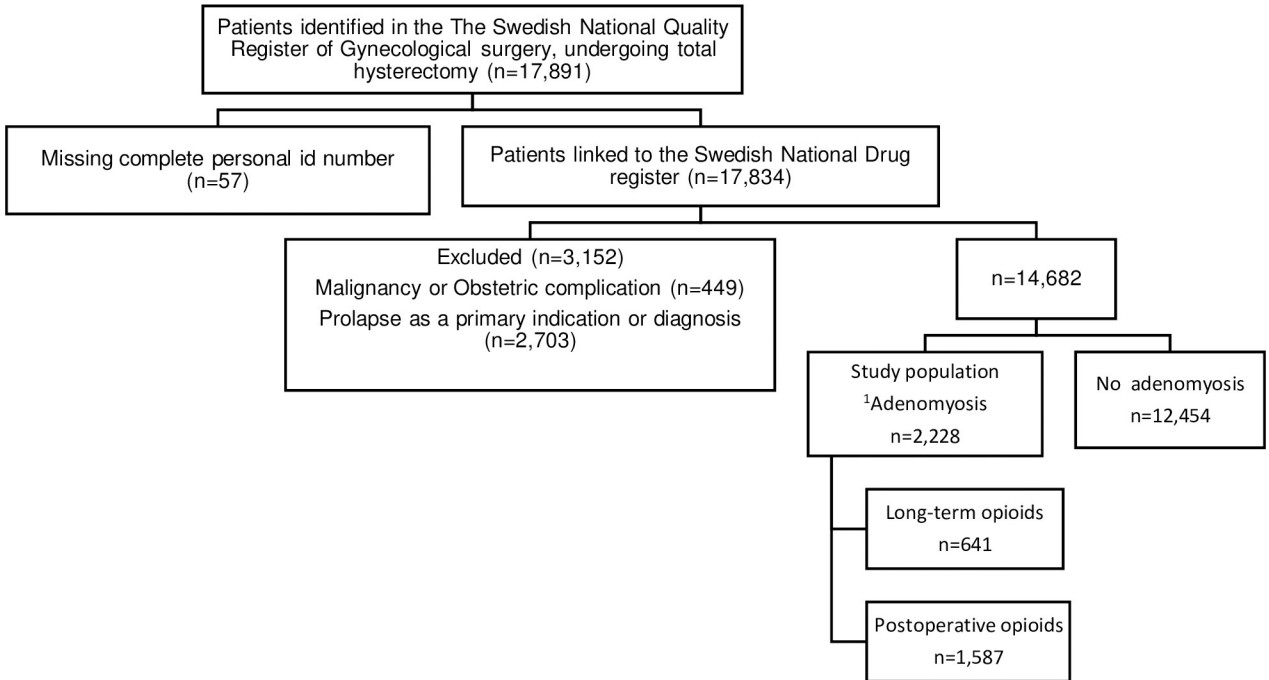

**Fig 1. Flowchart of the study population.** n = frequency (proportions %). Preoperative = one year preoperative. Long-term opioids = patients prescribed opioids > one year after surgery. Postoperative opioids = patients prescribed opioids < one year after surgery. [1]Postoperative diagnosis adenomyosis.

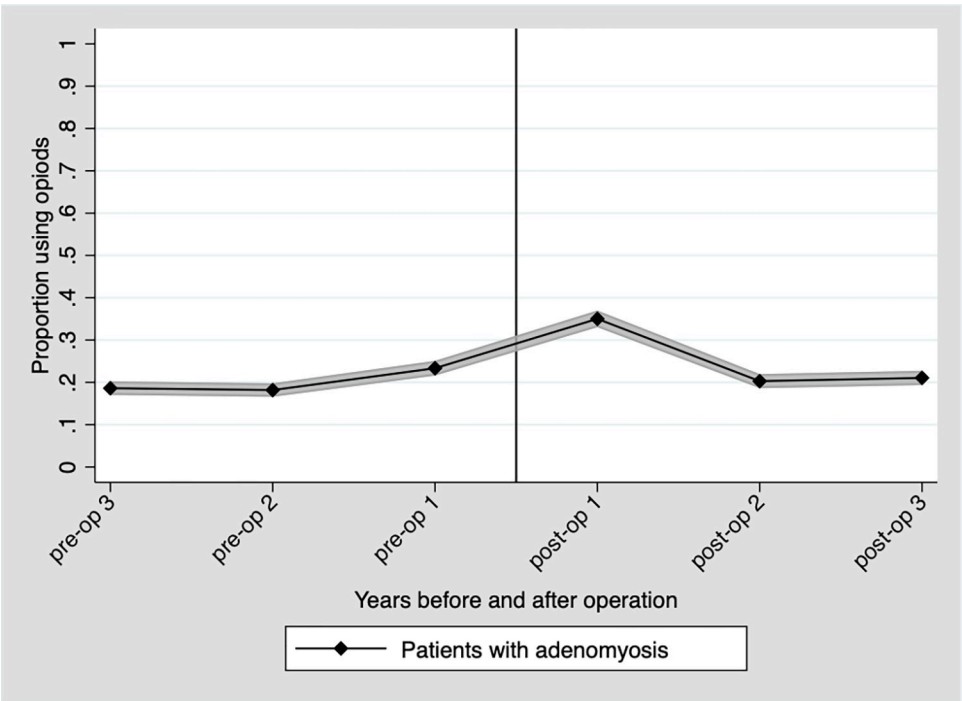

**Fig 2. Comparing frequency of women with adenomyosis (n = 2,228) using prescribed opioids three years before to three years after hysterectomy (Table 2).**

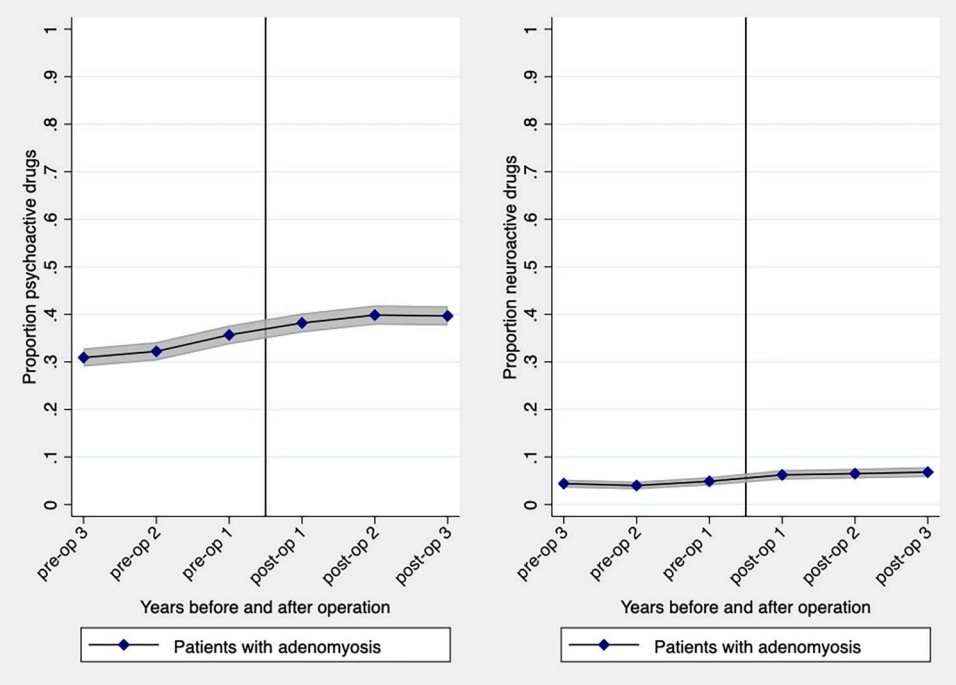

**Fig 3. Comparing frequency of women with adenomyosis (n = 2,228) using prescribed psycho- and neuroactive drugs three years before to three years after hysterectomy (Table 3).**

**Table 2.**

| Adenomyosis | [1]3 years before hysterectomy | 3 years after hysterectomy | OR (95% CI) |
|---|---|---|---|
| **Opioids** | [1]415 (18.6) | 469 (21.1) | 1.2 (1.01–1.3) |
| Non-opioids | [1]820 (36.8) | 903 (40.5) | 1.2 (1.0–1.3) |
| Analgesics overall | [1]921 (41.3) | 986 (44.3) | 1.2 (1.1–1.3) |

Graph showing point estimate and 95% confidence interval (shadowed area).

Preop = Before hysterectomy; Postop = After hysterectomy

[1]Reference

Data are presented as frequencies (proportions)

OR = odds ratio, CI = confidence interval

Non opioids = Paracetamol, Aspirin, Non steroid anti-inflammatory drugs

**Table 3.**

| Adenomyosis | [1]3 years before hysterectomy | 3 years after hysterectomy | OR 95% CI |
|---|---|---|---|
| Psychoactive drugs | [1]689 (30.9) | 884 (39.7) | 1.5 (1.3–1.7) |
| Neuroactive drugs | [1]109 (4.9) | 152 (6.8) | 1.6 (1.2–2.1) |

Graph showing point estimate and 95% confidence interval (shadowed area).

Preop = Before hysterectomy; Postop = After hysterectomy

[1]Reference

Data are presented as frequencies (proportions)

OR, Odds Ratio; 95% CI, 95% Confidence Interval

Non opioids = Paracetamol, Aspirin, Non steroid anti-inflammatory drugs

Perioperative complications occurred in 2.0% in the group of long-term opioid users, compared to 3.3% in the group of no long-term opioid users (aOR 0.7, 95% CI 0.3–1.3)).

## Discussion

This study shows that hysterectomy was not associated with a post-operative reduced opioid use in a population of patients with adenomyosis. Patients with preoperative use of opioids/antidepressants and uterine size <300g were at increased risk for long-term opioid use.

Of all women in Sweden aged 40–45 years during the same time-period as the study (2015–2018), 1.7–1.9% retrieved opioids [11]. Among patients with adenomyosis, long-term post-operative use of opioids is significantly elevated compared to all women undergoing hysterectomy; 21.1% vs 7.1% [7]. This finding is contradictory to a previous study on patients post hysterectomy (n = 443) where adenomyosis patients (n = 171) expressed less post-operative pain by self-reported questionnaires, compared to patients with no adenomyosis [12].

In this study we establish that pre-operative use of opioid/antidepressants is a risk factor of post-operative opioid use also in adenomyosis patients. This has previously been shown for other patient groups [13]. The finding of a smaller uterine size <300g as a risk factor, was unexpected. The reasons for these patients requiring opioids to a higher extent post-operatively remains unexplained. It is known however, that chronic pain includes several complex mechanisms including a correlation between symptom severity and duration of symptoms. This may suggest that the disease is aggravated over time, and is potent enough to engage pain circuits in the whole pelvis, not only the uterus [14, 15], which may explain remaining chronic pain conditions in some patients. We acknowledge the lack of systematic endometriosis differentiation at the time of the study (i.e the #Enzian score, which is since June 2023 a mandatory part of The Swedish National Quality Register of Gynecological Surgery). Although this is a weakness of this investigation, we consider the risk of undiagnosed severe endometriosis similar in patients with or without long-term opiod treatment, since the frequency of co-current endometriosis was almost exactly the same in both groups. The presence of non-gynecological conditions causing chronic pain (and subsequent use of opioids) could furthermore not be investigated in these registers.

The prevalence of adenomyosis among hysterectomized patients was 15%, a low proportion compared to for example Chinese data of 43% [16] and the Norwegian Adenomyosis Study prevalence of 30–63% [17, 18]. The latter studies were however single-site hysterectomy cohorts, whereas our study describe a national cohort with coverage of 93%. The Norwegian Adenomyosis Study also included a more thorough pathology protocol (including 5mm-slices of uterus, which is comparable to the Swedish protocol for suspected malign uterine disease) where the gynecologist responsible for the pre-operative diagnose also participated, which maximized the sensitivity (15), but does not represent the clinical setting in most countries, including ours.

It is plausible that the low level of correct pre-operative diagnose could have aided the development of chronic pain due to an undertreated condition, but our data do show that a vast majority (63.8–69.6%) did have pre-operative hormonal prescriptions, regarded as first line treatment för HMB/dysmenorrhea. Considering the well-known ongoing opioid pandemic, an opioid usage of approx. 20% in this otherwise healthy group (>96% ASA 1–2) deserves attention.

This study has several limitations.

Firstly, the symptomatology indicating surgery is not a mandatory question in the register, and the proportion of missing data was high. The registered data on Long-acting Reversible Contraceptives (LARC) is limited to prescription only, thus we cannot conclude when

discontinuation of LARC occurred. Furthermore, the missing data from patient-reported questionnaires was too high to include analysis of subjective pain levels pre- and post-operatively, which resulted in the analysis of opioid use as a proxy for pain. Therefore, we cannot conclude that the opioid use is due to pain caused by adenomyosis, but simply acknowledge that the use does not decline after hysterectomy.

Although the effect of hysterectomy treatment on heavy menstrual bleeding is curative, our data suggests that hysterectomy does not reduce need for opioid treatment among adenomyosis patients. However, according to what we discuss above, we cannot conclude that hysterectomy does not reduce subjective pain levels in patients with adenomyosis. Further research into which women benefit most from surgical intervention with hysterectomy for adenomyosis with a special focus on pre- and perioperative diagnosis of deep endometriosis should be a focus to improve patient counseling.

## Conclusion

Hysterectomy does not reduce long-term use of opioids post-surgery in patients with adenomyosis in this study. Subjective reduction of postoperative pain was not possible to investigate due to large amount of missing data. Risk-factors for continuous use of opioids after hysterectomy in patients with adenomyosis is pre-operative use of opioids/antidepressants and uterine size <300g.

## Supporting information

**S1 Fig. A comparison of World Health Defined Daily Dose prescribed opioids in patients with adenomyosis (n = 2,228) three years before to three years after hysterectomy.** y-axis: frequency of World Health Organization Defined Daily Dose. x-axis: preop = preoperatively, postop = postoperatively. Patients with adenomyosis. 3 years before compared to 3 years after hysterectomy (IRR 1.1 95% CI 1.0–1.3).
(TIF)

**S2 Fig. Comparing frequency of women with adenomyosis (n = 2,228) using prescribed psycho- and neuroactive drugs three years before to three years after surgery in different routes of hysterectomy.** Analyze of interaction in proportion of opioid users including hysterectomy procedure and study year (base outcome vaginal hysterectomy and 3 years preoperative). Graph showing point estimate and 95% confidence interval (shadowed area). TLH/RATLH = Laparoscopic/Robotic assisted hysterectomy AH = Abdominal hysterectomy VH = Vaginal hysterectomy. Preop = Before hysterectomy; Postop = After hysterectomy. OR, Odds Ratio; 95% CI, 95% Confidence Interval. a = adjusted for baseline data including all variables from Table 1.
(TIF)

**S3 Fig. Comparing frequency of women with adenomyosis (n = 2,228) and pure adenomyosis (n = 713) using prescribed opioids three years before to three years after hysterectomy.** Graph showing point estimate and 95% confidence interval (shadowed area). Preop = Before hysterectomy; Postop = After hysterectomy. [1]Reference. Data are presented as frequencies (proportions). OR = odds ratio, CI = confidence interval. Non opioids = Paracetamol, Aspirin, non steroid anti-inflammatory drugs. *Pure adenomyosis = only patients with post-operative pathology main diagnoses of adenomyosis (co-diagnoses adenomyosis excluded) + any diagnosis of endometriosis was excluded.
(TIF)

**S1 Table. A description of how drugs were grouped from the Swedish Prescribed Drug Register.**
(DOCX)

**S2 Table. S2 Fig data.**
(DOCX)

## Acknowledgments

Biostatistician Henrike Häbel, PhD, Institute for Environmental Medicine, Karolinska Institutet, Stockholm, Sweden. No conflicts of interest to declare.
Biostatistician Anna Warnqvist, PhD, Institute for Environmental Medicine, Karolinska Institutet, Stockholm, Sweden. No conflicts of interest to declare.

## Author Contributions

**Conceptualization:** Malin Brunes, Marion Ek, Klara Hasselrot.

**Data curation:** Malin Brunes.

**Formal analysis:** Malin Brunes.

**Funding acquisition:** Malin Brunes, Marion Ek, Klara Hasselrot.

**Investigation:** Malin Brunes, Klara Hasselrot.

**Methodology:** Malin Brunes, Marion Ek, Klara Hasselrot.

**Supervision:** Marion Ek, Klara Hasselrot.

**Validation:** Klara Hasselrot.

**Writing – original draft:** Malin Brunes, Gudny Jonsdottir, Klara Hasselrot.

**Writing – review & editing:** Malin Brunes, Gudny Jonsdottir, Marion Ek, Helena Kopp Kallner, Klara Hasselrot.

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
