## [Decision Letter · Decision Letter 0]

19 Aug 2024

PONE-D-24-18453Impact of hysterectomy on opioid use in patients with adenomyosis: a nationwide register studyPLOS ONE

Dear Dr. Brunes,

Thank you for submitting your manuscript to PLOS ONE. After careful consideration, we feel that it has merit but does not fully meet PLOS ONE’s publication criteria as it currently stands. Therefore, we invite you to submit a revised version of the manuscript that addresses the points raised during the review process.

We look forward to receiving your revised manuscript.

Kind regards,

Kazunori Nagasaka

Academic Editor

PLOS ONE

Journal Requirements:

"Malin Brunes have received honoraria from Intuitive Surgical for three lectures and proctoring."

Additional Editor Comments:

Dear Authors,

Thank ypu for your submission to Plos One.

Please find the reviewer's comments and consider their suggestion.

We look forward to your revised manuscript soon.

Sincerely,

Kazunori Nagasaka

Reviewers' comments:

Reviewer's Responses to Questions

**Comments to the Author**

1. Is the manuscript technically sound, and do the data support the conclusions?

Reviewer #1: Partly

Reviewer #2: Partly

2. Has the statistical analysis been performed appropriately and rigorously? 

Reviewer #1: Yes

Reviewer #2: I Don't Know

3. Have the authors made all data underlying the findings in their manuscript fully available?

Reviewer #1: Yes

Reviewer #2: No

4. Is the manuscript presented in an intelligible fashion and written in standard English?

Reviewer #1: Yes

Reviewer #2: Yes

5. Review Comments to the Author

Reviewer #1: Dear authors, please find my comments to your manuscript here:

- please comment on the fact that only 3.9 % of patients with adenomyosis have had an indication for hysterectomy because of adenomyosis. Did you analyze the indication in the other 96.1 % of patients with adenomyosis?

- In Table 1 you added info on additional Endometriosis. But there is no differentiation of peritoneal and deep endometriosis and the type and localization. Do you think that a reason for continuous opioid use could be underdiagnosed and therefore undertreated DE? Missed bowel endometriosis? etc.

- Please comment on the possible lack of detection of additional deep endometriosis in the time of the study (2012-2015) (there current role of pre surgical imaging was less common at that time).

- Line 302: please explain your speculation. In my opinion there is no clear explanation for your assumption. If you have no reference, this part should be omitted.

- Line 311: as mentioned above, this should be discussed in detail.

- Line 315: please explain if a guideline for histopathological workup of hysterectomy specimens in search for adenomyosis exists in Sweden? Would you think it's possible, that the proportion of adenomyosis could be much higher, because pathologists missed the disease? Please discuss.

- Your conclusion gives the impression that hysterectomy does not decrease dysmenorhoe and pain in patients with adenomyosis. In my opinion, your results can not support this assumption and therefore this fact should be made very clear in the conclusion and the abstract.

- In my opinion, the most important limitation of your work is the fact, that you look at adenomyosis without looking at endometriosis. Of course, adenomyosis can be found without additional peritoneal or deep endometriosis, but in many patients (references) both conditions can be found simultaneously. This should be discussed.

Best regards.

Reviewer #2: Thank you for this clear manuscript of an important issue.

ABSTRACT

Methods: retrospective/historical register-based study?

Results: lines 51-54: .. preoperative use of opioids and antidepressants

Conclusions: Should be more in line with the results. Hysterectomy did not reduce … in our study. .. However, other conditions affecting opioid use were not known.

INTRODUCTION

Page 5, line 77: “>160 years later “– rather “more than 160 later..”?

MATERIAL AND METHODS

- I would like to know all medications you have retrieved to your study. I suggest you add a supplementary table including all ATC codes and affecting agents you have included in this study.

- Determine which opioids were included (mild and strong as well?)

- You mention muscle-relaxants here as well, were they included somewhere?

- You do not mention where you have retrieved the hormonal treatment before – patient of gynecologist based data from register or from SPDR?

- Do you know how many of the women had levonorgestrel releasing intrauterine device preoperatively (specific ATC code)?

- Did all have three year pre and post operative follow-up data surveillance with medicines? Did you have surveillance of emigrations, death – was this assessed?

- How the DDD was calculated in your study? All bought medicines in one years/365 days? Define this clearly.

- The subanalyses (for potential risk factor to lon-term use) would have been concomitant endometriosis/solely adenomyosis since many might stop their hormonal medication after surgery

- Did you assess those whose indication was adenomyosis preoperatively separately (only 86)?

- Exclusion criterias are presented here – prolapses excluded, why?

RESULTS

- The similarity on number of endometriosis and hormonal preoper treatment between the long-users and non-users coud be mentioned here (Table 1)

- Please mention here n(%) of the types of surgeries (AH, TLH, RTLH, VH)

- How many of the used mild, how many strong opioids before and after?

- The use of other drugs in refs (in women not using opioids)?

- In which age group the long term opioids used the most?

- Unfortunately watching

o Figure 2 The Table under OR 1.2 (95%CI 1.0-1.3 : I do not find this significantly higher, the overall use

o Figure S1 and IRR 1.1 (95%CI 1.0-1.3) I do not find the latter time DDD significantly higher but not lower either.

DISCUSSION

- Rather “This study shows that the hysterectomy was not associated with…”

- How many of the women are symptomless with adenomyosis? This might be discussed as well.

FIGURES AND TABLES

Difficult to understand as partly in text and partly in the end!

Figure 1 – flowchart: dark

Figure 2- Was the point estimate of persons using in preop 3 years before or the mean of preop 3 years- but up to preop 2 years (should be in methods how calculated)

If mean - your dots in the figure might be better in the middle of two time points?

The table under this is not clear: Adenomyosis vs years? The references, where, should be clarified! Figure 2 is before Table 1 – the adjusted variables has to listed here

Figure 3- These figures could be joined to one.

Figure S2- “…hysterectomy+years”? Explanations for THL/RATLH etc lacking

Table 1 Adjusted variables should be

6. PLOS authors have the option to publish the peer review history of their article (what does this mean?). If published, this will include your full peer review and any attached files.

Reviewer #1: No

Reviewer #2: No

---

## [Author Response · Author response to Decision Letter 0]

13 Sep 2024

Dear Editor,

We are pleased that the reviewers helped us to improve this manuscript through their valuable suggestions, and that you consider our article for publication in your journal. We have read and appreciated the comments from the reviewers, please see our responses below.

Response to Reviewers 

Dear reviewers,

We are very grateful for your valuable comments that really have helped us improve our paper.

Reviewer #1: Dear authors, please find my comments to your manuscript here:

1.- please comment on the fact that only 3.9 % of patients with adenomyosis have had an indication for hysterectomy because of adenomyosis. Did you analyze the indication in the other 96.1 % of patients with adenomyosis?

Thank you for a very relevant remark. In Sweden, the indication for the surgical procedure is identified as the diagnosis the patient received on the day of surgery. What we mean here, is that only 3,9% of patients who had a hysterectomy had a main diagnosis of adenomyosis on the day of the hysterectomy (other adenomyosis patients might have had “menorraghia” as main diagnosis, for example). 

Firstly, we believe that the low rate of correct pre-op diagnosis is worth highlighting due to the (hopefully, former) low awareness of adenomyosis and the corresponding signs on TVUS. 

Secondarily, many patients are scheduled for hysterectomy in Sweden with symptom-related diagnosis only, where adenomyosis might be the underlying cause, for example ”menorraghia”(36%) or ”dysmenorrhea”(1.5%). Also the large group of ”myoma” (18.9%) might include misdiagnosed adenomyosis/adenomyoma, and 30-40% of the adenomyosis patients did indeed also have myomas and/or endometriosis (2,7% of total indications). 29,5% of indications were unfortunately missing. 

We have now added this information in Results section for clarification, as below (line 216-219:

Before surgery, 86 of 2,228 patients (3.9%) had adenomyosis registered as main indication for surgery. Main registered indications for hysterectomy (on the day of surgery) in this dataset includes “menorraghia”(36%), “myoma” (18.9%), “dysplasia”(2.8%), “adnexal pathology”(2.7%), “endometriosis”(2.7%) and “dysmenorrhea”(1.5%), and the rate of missing data was 29.5%.

2.- In Table 1 you added info on additional Endometriosis. But there is no differentiation of peritoneal and deep endometriosis and the type and localization. Do you think that a reason for continuous opioid use could be underdiagnosed and therefore undertreated DE? Missed bowel endometriosis? etc. 

Thank you for this valid point. The reason of the paucity of endometriosis differentiation, is that the #Enzian score (which is now mandatory part of the register) was not established at the time of the study. In the dataset presented here, the amount of missing data on endometriosis differentiation was 43% and 47% in the two groups of adenomyosis patients, a level we consider too high to use in an analysis. There is however a statistically low risk of significantly underdiagnosed deep endometriosis in the long-term group compared to the no long-term group, since the proportion of co-current endometriosis is almost exactly the same in the two groups (15,4% and 15,9%, respectively). This has now been clarified in the Result section, as below (line 316-318):

Neither parity, BMI, smoking or previous abdominal surgery was significantly associated with long-term postoperative consumption of opioids, neither was the prevalence of endometriosis (15.4% in long-term users vs 15.9%) or the use of preoperative hormonal treatment (69.6% in long-term users vs 63.8%) (table 1).

And in Discussion section (line 367-379):

We acknowledge the lack of systematic endometriosis differentiation at the time of the study (i.e the #Enzian score, which is since June 2023 a mandatory part of The Swedish National Quality Register of Gynecological Surgery). Although this is a weakness of this investigation, we consider the risk of undiagnosed deep endometriosis similar in patients with or without long-term opiod treatment, since the frequency of co-current endometriosis was almost exactly the same in both groups.

-3. Please comment on the possible lack of detection of additional deep endometriosis in the time of the study (2012-2015) (there current role of pre surgical imaging was less common at that time).

It is definitely also our experience, that pre-surgical imaging was less systematic regarding endometriosis in the years of the study. To aid us in future studies, we have now added the #Enzian score of endometriosis as mandatory in the national register of gynecological surgery. We are hoping for better pre-surgical imaging also of adenomyosis (as it is also included in the #Enzian score) in the future. Please see Discussion section for additional comments (stated in question 2 above). 

-4. Line 302: please explain your speculation. In my opinion there is no clear explanation for your assumption. If you have no reference, this part should be omitted.

We understand your remark, and have omitted this speculation as suggested.

-5. Line 311: as mentioned above, this should be discussed in detail. 

Thank you for this remark. The reference of endometriosis prevalence is from the register, however we have now extended the discussion, mentioning the lack of detailed endometriosis diagnostics at the time of study. Please see response to questions 2 for corresponding changes in the manuscript. 

6.- Line 315: please explain if a guideline for histopathological workup of hysterectomy specimens in search for adenomyosis exists in Sweden? Would you think it's possible, that the proportion of adenomyosis could be much higher, because pathologists missed the disease? Please discuss.

We totally agree that the low level of detected adenomyosis is due to low awareness – but it might also be due to the operating gynecologist, and not only the pathologist – if the surgeon has a low awareness and do not ask for presence of adenomyosis, the chance of a thorough histopathology examination in this regard is probably lower. In Sweden all uteri with suspected benign disease are examined macroscopically by the pathologist, followed by 2 biopsies from cervix uteri and 4 biopsies from corpus uteri, and additional biopsies from myomas >5cm. From macroscopically suspected areas (such as overt adenomyosis) additional biopsies are taken. Only in suspected malign disease, 5mm slices are used. 

We have added a comment on the pathology routine in the Material&Methods as well as in the Discussion section, please see below:

M&M line 132-135: In Sweden all uteri with suspected benign disease are examined macroscopically by the pathologist, followed by 2 biopsies from cervix uteri and 4 biopsies from corpus uteri, and additional biopsies from myomas >5cm. From macroscopically suspected areas (such as overt adenomyosis) additional biopsies are taken

.

Discussion line 387-391: The Norwegian Adenomyosis Study also included a more thorough pathology protocol (including 5mm-slices of uterus, which is comparable to the Swedish protocol for suspected malign uterine disease) where the gynecologist responsible for the pre-operative diagnose also participated, which maximized the sensitivity (15), but does not represent the clinical setting in most countries, including ours. 

7.- Your conclusion gives the impression that hysterectomy does not decrease dysmenorhoe and pain in patients with adenomyosis. In my opinion, your results can not support this assumption and therefore this fact should be made very clear in the conclusion and the abstract.

Thank you for this remark. We have now clarified that opioid use is a proxy for pain and other conditions affecting opioid use is unknown. We have furthermore clarified the lack of data on self-reported pain post-operatively, which is of course a weakness of the study. 

We have now made clarifications in the abstract, the Discussion and in the Conclusion section, as shown below:

Abstract line 67: Hysterectomy does not reduce use of opioids among adenomyosis patients in long-term follow up, although other conditions affecting opioid use were not known.

Discussion line 347-348: This study shows that hysterectomy was not associated with a post-operative reduced opioid use in a population of patients with adenomyosis.

line 405-407

Furthermore, the missing data from patient-reported questionnaires was too high to include analysis of subjective pain levels pre- and post-operatively, which resulted in the analysis of opioid use as a proxy for pain. Therefore, we cannot conclude that the opioid use is due to pain caused by adenomyosis, but simply acknowledge that the use does not decline after hysterectomy.

Conclusion line 418: Hysterectomy does not reduce use of opioids post-surgery in patients with adenomyosis, although other condition affecting opioid use was not known.

8. In my opinion, the most important limitation of your work is the fact, that you look at adenomyosis without looking at endometriosis. Of course, adenomyosis can be found without additional peritoneal or deep endometriosis, but in many patients (references) both conditions can be found simultaneously. This should be discussed.

We absolutely agree with the reviewer that the differentiation of endometriosis is clearly suboptimal in this dataset, as we added to the discussion as suggested above (see question 2 and 3). The #Enzian score will hopefully aid better research in the future. To clarify, we have added to the Result section the similar levels of co-existing endometriosis diagnoses in the two adenomyosis groups (15.4% and 15.9%, respectively) as stated in question no 2. 

To further address the possible characteristics of the group of adenomyosis only (here described as “pure adenomyosis”, vs adenomyosis+concomitant condition such as endometriosis and/or fibroids) we have added a supplementary figure, where it is shown that there was no significant statistical difference between these two groups, regarding levels of opioid use pre- and post surgery:

This figure is added in the Result section together with the following text, to address this remark (line 324-327) :

To address possible characteristics of adenomyosis only (no concomitant endometriosis and/or myomas) vs adenomyosis as additional diagnosis, we performed a subgroup analysis (fig S3) showing no significant difference in opioid use between these two groups.

Reviewer #2: 

Thank you for this clear manuscript of an important issue.

Thank you very much for that comment!

ABSTRACT

1. Methods: retrospective/historical register-based study?

We agree with the reviewer that this is a retrospective register-based cohort study, and this is now clarified in the abstract (line 52): This is a nationwide retrospective register-based cohort study

2. Results: lines 51-54: .. preoperative use of opioids and antidepressants

Conclusions: Should be more in line with the results. Hysterectomy did not reduce … in our study. .. However, other conditions affecting opioid use were not known.

Thank you for this remark, this has been clarified as suggested (line 67): 

Hysterectomy does not reduce use of opioids among adenomyosis patients in long-term follow up, although other conditions affecting opioid use were not known.

INTRODUCTION

3. Page 5, line 77: “>160 years later “– rather “more than 160 later..”? 

This is now corrected.

MATERIAL AND METHODS

4.- I would like to know all medications you have retrieved to your study. I suggest you add a supplementary table including all ATC codes and affecting agents you have included in this study.

Thank you for the valuable suggestion, the table suggested is now added (as below) in the Material&Methods section:

Supplementary table. A description of how drugs were grouped from the Swedish Prescribed Drug Register.

5.- Determine which opioids were included (mild and strong as well?) 

As shown in the table, all opioids were included.

6.- You mention muscle-relaxants here as well, were they included somewhere? 

Yes, as shown in the table. Muscle-relaxants were a (very small) proportion of the analgesics group, and were included here.

7. - You do not mention where you have retrieved the hormonal treatment before – patient of gynecologist based data from register or from SPDR? 

It comes from the SPDR, we believe this is now clarified when we added the table, again a very good suggestion.

8.- Do you know how many of the women had levonorgestrel releasing intrauterine device preoperatively (specific ATC code)?

Thank you for this important remark. Levonorgestrel releasing IUD was included in the hormonal treatment section, as clarified above in the newly inserted table: However, since data was retrieved from the SPDR we did not know if or when the hormonal IUD were discontinued (this is also the reason why we did not perform a subgroup analysis of this group). 

This is a clear weakness of this study and has been clarified under the suggested Supplementary Table 1 as well as in the Discussion section (line 403-405):

The registered data on Long-acting Reversible Contraceptives (LARC) is limited to prescription only, thus we cannot conclude when discontinuation of LARC occurred.

9.- Did all have three year pre and post operative follow-up data surveillance with medicines? Did you have surveillance of emigrations, death – was this assessed?

Yes, all patients have 3 years pre and post operative data on medications, they are retrieved electronically straight from all pharmacies to the register. We did not have data of emigrations and deaths, however the dataset consists of >96% individuals of ASA group 1-2, the numbers of deaths should thus be extremely low, and should be equal in both groups.

10.- How the DDD was calculated in your study? All bought medicines in one years/365 days? Define this clearly.

Thank you for this valuable comment. Data was collected yearly during the study period, this has been clarified in the Material&Methods section (DDD was thus calculated yearly) line 159-162: 

Data on the proportion of patients using the studied medications was collected for each year of the study period. As a secondary outcome, in the opioid group, differences in the World Health Organization Defined Daily Dose (DDD) were analyzed yearly before and after hysterectomy

11.- The subanalyses (for potential risk factor to long-term use) would have been concomitant endometriosis/solely adenomyosis since many might stop their hormonal medication after surgery

Thanks for a very interesting remark. Surely, on can suspect adenomyosis patients to discontinue hormonal medications postop to a higher extent than patients with combined adeno/endo. Due to our curiosity regarding possible characteristics of the “pure” adenomyosis group (no concomitant endometriosis and/or myomas) when we started the study, a subgroup analysis of this group has previously been made. We did however not include this figure in the original manuscript, since there were no apparent differences between opioid use over time in pure adenomyosis patients vs adeno patients with combined disease. We have now once again included this figure as Supplementary (fig S3), to partly answer your (and probably others) question, together with the following text in the Results section (line 324-327):

To address possible characteristics of adenomyosis only patients (no concomitant endometriosis and/or myomas) vs adenomyosis as additional diagnosis, we performed a subgroup analysis (S3 Fig) showing no significant difference in opioid use between these two groups.

12.- Did you assess those whose indication was adenomyosis preoperatively separately (only 86)? 

No, the ethical permission did not allow us to extract individual patient records.

13.- Exclusion criterias are presented here – prolapses excluded, why? 

Prolapse consisted of only 15% of all hysterectomies during the study period, and concerns mainly post menopausal patients. Since adenomyosis rarely is symptomatic post menopaus, we chose to exclude this indication.

RESULTS

14.- The similarity on number of endometriosis and hormonal preoper treatment between the long-users an

---

## [Decision Letter · Decision Letter 1]

4 Nov 2024

PONE-D-24-18453R1Impact of hysterectomy on opioid use in patients with adenomyosis: a nationwide register studyPLOS ONE

Dear Dr. Brunes,

Thank you for submitting your manuscript to PLOS ONE. After careful consideration, we feel that it has merit but does not fully meet PLOS ONE’s publication criteria as it currently stands. Therefore, we invite you to submit a revised version of the manuscript that addresses the points raised during the review process.

We look forward to receiving your revised manuscript.

Kind regards,

Kazunori Nagasaka

Academic Editor

PLOS ONE

Journal Requirements:

Additional Editor Comments:

Dear Authors,

We sincerely apologize for the delay. The peer review has been completed, and we will share it with you shortly.

The points raised by the two reviewers are highly relevant, particularly regarding the presence of deep endometriosis in some patients experiencing prolonged pain,

such as back pain and dyspareunia, after total hysterectomy for uterine adenomyosis.

Please revise the text according to the reviewers' remarks.

We look forward to receiving your revised paper.

I believe your research is excellent and will significantly contribute to obstetric and gynecological practice.

Sincerely,

Plos One

Kazunori Nagasaka

Reviewers' comments:

Reviewer's Responses to Questions

**Comments to the Author**

1. If the authors have adequately addressed your comments raised in a previous round of review and you feel that this manuscript is now acceptable for publication, you may indicate that here to bypass the “Comments to the Author” section, enter your conflict of interest statement in the “Confidential to Editor” section, and submit your "Accept" recommendation.

Reviewer #1: All comments have been addressed

Reviewer #2: All comments have been addressed

2. Is the manuscript technically sound, and do the data support the conclusions?

Reviewer #1: Partly

Reviewer #2: Yes

3. Has the statistical analysis been performed appropriately and rigorously? 

Reviewer #1: Yes

Reviewer #2: Yes

4. Have the authors made all data underlying the findings in their manuscript fully available?

Reviewer #1: Yes

Reviewer #2: Yes

5. Is the manuscript presented in an intelligible fashion and written in standard English?

Reviewer #1: Yes

Reviewer #2: Yes

6. Review Comments to the Author

Reviewer #1: Dear authors,

thank you for addressing all comments. Although you added additional information and your work is very good and very readable, I am still not fully convinced regarding your conclusion. Please let my explain my doubts:

It reads as if hysterectomy does not help patients in case of adenomyosis. There is relevant literature that shows hysterectomy to be an efficient treatment of adenomyosis related symptoms. Publications also show that patients with deep endometriosis and adenomyosis have a higher risk of re-intervention in case of previous fertility sparing deep endo excision surgery due to persistent adenomyosis symptoms.

The fact that patients with adenomyosis might also suffer from additional under diagnosed deep endometriosis should be further discussed. Post-hysterectomy deep endometriosis is a problematic condition that could be additionally explained.

Another aspect is the opioid use itself. Why would almost 20 % of patients use opioids pre-surgically? What's the reason for that? And does the addictional aspect of opioids add to ongoing use?

In my opinion, it would be important to add these open questions to discuss to the conclusion and abstract. Many readers go quickly through these sections and in this case would learn: hysterectomy does not seem to work in adenomyosis related pain. And in my opinion this should not be the message of your study. Especially as the data is some years old and things changed in the meantime.

Early diagnosis, better TVS quality, less opioids, complete adenomyosis and endometriosis excision....that would make the difference.

Best regards.

Reviewer #2: Thank you. My questions were explained and clarified well.

I find this study important to be published.

However, jus a few minor revisions:

-I would add to Results that women aged 45-54 were the long term users who used most opioids.

-In addition, in Introduction I would spell out the most used opioids in Sweden (parasetamol plus codeine, tramadol?) and I would consider placing the most important opioid drugs by their general names in the Methods section.

7. PLOS authors have the option to publish the peer review history of their article (what does this mean?). If published, this will include your full peer review and any attached files.

Reviewer #1: No

Reviewer #2: No

---

## [Author Response · Author response to Decision Letter 1]

11 Nov 2024

Response to Reviewers

Dear Reviewers,

Thank you for your interest in this important subject and your valuable comments helping us improve this manuscript

Reviewer #1: Dear authors,

thank you for addressing all comments. Although you added additional information and your work is very good and very readable, I am still not fully convinced regarding your conclusion. Please let my explain my doubts:

It reads as if hysterectomy does not help patients in case of adenomyosis. There is relevant literature that shows hysterectomy to be an efficient treatment of adenomyosis related symptoms. Publications also show that patients with deep endometriosis and adenomyosis have a higher risk of re-intervention in case of previous fertility sparing deep endo excision surgery due to persistent adenomyosis symptoms.

We agree that our conclusion is not in line with some previous publications, this is discussed in line 327-332.

The fact that patients with adenomyosis might also suffer from additional under diagnosed deep endometriosis should be further discussed. Post-hysterectomy deep endometriosis is a problematic condition that could be additionally explained.

We fully agree that undiagnosed presence of deep endometriosis could be a reason of continuous pain after hysterectomy, and have now clarified the Conclusions, both in the Abstract Line 58-60: 

Hysterectomy does not reduce opioid use among adenomyosis patients in long-term follow-up, although the subjective reduction of pain was not investigated in this study. Women with preoperative use of opioids/antidepressants and uterine size <300g are at increased risk for chronic opioid use. 

 as well as in the main manuscript, line 396-401:

Hysterectomy does not reduce long-term use of opioids post-surgery in patients with adenomyosis in this study. Subjective reduction of postoperative pain was not possible to investigate due to large amount of missing data. Risk-factors for continuous use of opioids after hysterectomy in patients with adenomyosis is pre-operative use of opioids/antidepressants and uterine size <300g. 

and in discussion line 384-391:

Although the effect of hysterectomy treatment on heavy menstrual bleeding is curative, our data suggests that hysterectomy does not reduce need for opioid treatment among adenomyosis patients. However, according to what we discuss above, we cannot conclude that hysterectomy does not reduce subjective pain levels in patients with adenomyosis. Further research into which women benefit most from surgical intervention with hysterectomy for adenomyosis with a special focus on pre- and perioperative diagnosis of deep endometriosis should be a focus to improve patient counseling.

Another aspect is the opioid use itself. Why would almost 20 % of patients use opioids pre-surgically? What's the reason for that? And does the addictional aspect of opioids add to ongoing use?

We believe that the high proportion of opioid users is due to the high prevalence of chronic pain in this patient group, as well as an effect of the well-known overprescription of opioids as a treatment for chronic pain. This is discussed in line 365-370

In my opinion, it would be important to add these open questions to discuss to the conclusion and abstract. Many readers go quickly through these sections and in this case would learn: hysterectomy does not seem to work in adenomyosis related pain. And in my opinion this should not be the message of your study. Especially as the data is some years old and things changed in the meantime.

Early diagnosis, better TVS quality, less opioids, complete adenomyosis and endometriosis excision....that would make the difference.

We totally agree, and hope for: better diagnostics, carefully chosen surgical interventions and lesser opioids in the future, regarding this unfortunate patient group. This has been clarified in discussion line 384-391:

Although the effect of hysterectomy treatment on heavy menstrual bleeding is curative, our data suggests that hysterectomy does not reduce need for opioid treatment among adenomyosis patients. However, according to what we discuss above, we cannot conclude that hysterectomy does not reduce subjective pain levels in patients with adenomyosis. Further research into which women benefit most from surgical intervention with hysterectomy for adenomyosis with a special focus on pre- and perioperative diagnosis of deep endometriosis should be a focus to improve patient counseling.

Best regards.

Reviewer #2: Thank you. My questions were explained and clarified well.

I find this study important to be published.

However, jus a few minor revisions:

-I would add to Results that women aged 45-54 were the long term users who used most opioids.

We are sorry, but there was a mistake it was the age group < 45 that was the most common age group in long term users 314 (49 %) of the patients in the long term opioid group were <45 years old. However, age was not a statistically significant risk factor for long term use.

As suggested, this has been clarified in results line 271-273:

A total of 314 patients (49%) in the long-term opioid group were under 45 years old. However, age was not identified as a statistically significant risk factor for long-term opioid use.

-In addition, in Introduction I would spell out the most used opioids in Sweden (parasetamol plus codeine, tramadol?) and I would consider placing the most important opioid drugs by their general names in the Methods section.

Thank you for this valuable comment. This has been added in Introduction line 97-99:

The most commonly used opioids in Sweden from 2012 to 2015 were oxycodone, tramadol and paracetamol combined with codeine. 

Ref 9: Förskrivning av opioider i Sverige

Läkemdel doser och diagnoser (Prescription of opioids in Sweden, Drugs, doses and diagnoses.) Swedish medical products academy; 2020-02-17 [cited 2020-02-17]. Available from: https://www.lakemedelsverket.se/globalassets/dokument/publikationer/lakemedelsprodukter-och-narkotika/forskrivning-av-opioider-i-sverige-2020-1.pdf

And in Methods line 140-142:

ATC code N02A (opioids) includes, for example, morphine, oxycodone and tramadol, as well as combination drugs containing paracetamol, along with codeine.

The reference list has been corrected with one updated website.

Reference list:

11. Socialstyrelsen. Läkemedelsregistret 2005. Available from: https://www.socialstyrelsen.se/statistik-och-data/register/lakemedelsregistret/.

---

## [Decision Letter · Decision Letter 2]

22 Dec 2024

Impact of hysterectomy on opioid use in patients with adenomyosis: a nationwide register study

PONE-D-24-18453R2

Dear Dr. Brunes,

We’re pleased to inform you that your manuscript has been judged scientifically suitable for publication and will be formally accepted for publication once it meets all outstanding technical requirements.

Kind regards,

Kazunori Nagasaka

Academic Editor

PLOS ONE

Additional Editor Comments: 

The research is very important in the feild. Now I think the manuscript is acceptable for publication.

Dear Authors,

Congratulations on your publication in PLOS One!

This is a great achievement, and I’m sure it will contribute significantly to the field.

Sincerely,

Kazunori Nagasaka

Reviewers' comments:

Reviewer's Responses to Questions

**Comments to the Author**

1. If the authors have adequately addressed your comments raised in a previous round of review and you feel that this manuscript is now acceptable for publication, you may indicate that here to bypass the “Comments to the Author” section, enter your conflict of interest statement in the “Confidential to Editor” section, and submit your "Accept" recommendation.

Reviewer #1: All comments have been addressed

Reviewer #3: (No Response)

2. Is the manuscript technically sound, and do the data support the conclusions?

Reviewer #1: Yes

Reviewer #3: Yes

3. Has the statistical analysis been performed appropriately and rigorously? 

Reviewer #1: Yes

Reviewer #3: I Don't Know

4. Have the authors made all data underlying the findings in their manuscript fully available?

Reviewer #1: Yes

Reviewer #3: Yes

5. Is the manuscript presented in an intelligible fashion and written in standard English?

Reviewer #1: Yes

Reviewer #3: No

6. Review Comments to the Author

Reviewer #1: Dear authors, thank you for addressing all comments. The additional explanations in the discussion clarify the analysis and therefore, I recommend publication of your work in its current form.

Best regards.

Reviewer #3: The statistics sound ok, but I am not a statistician.

This is my first review and I have various comments and suggestions, see my attachment.

Previous reviewer's comments have in principal been addressed.

7. PLOS authors have the option to publish the peer review history of their article (what does this mean?). If published, this will include your full peer review and any attached files.

Reviewer #1: No

Reviewer #3: No

---

## [Editor Report · Acceptance letter]

26 Dec 2024

PONE-D-24-18453R2 

PLOS ONE

Dear Dr. Brunes, 

I'm pleased to inform you that your manuscript has been deemed suitable for publication in PLOS ONE. Congratulations! Your manuscript is now being handed over to our production team.

Kind regards, 

on behalf of

Professor Kazunori Nagasaka 

Academic Editor

PLOS ONE